# Theoretical morphospace reveals mixed optimisation of the avian wing planform for flight style

Benton Walters ⬤ ✉, Yuming Liu, Emily J. Rayfield ⬤ ✉ &
Philip C. J. Donoghue ⬤ ✉

Bird wings exhibit a broad degree of functional and shape variation, though the exact nature of the form-function relationship is uncertain. Recent analysis suggests that functional variability is explained by linear non-shape-based traits and that shape variation is largely explained by phylogeny. We assay the relationship between wing planform shape and functional performance using a theoretical morphospace approach that eschews assumptions of the functional optimality of empirical morphologies. Hypothesised empirical properties are considered post hoc relative to their positions in performance surfaces. We produce a theoretical morphospace of wing planform shape and deduce the functional performance and optimality of 1139 extant taxa. Functional tests cover metrics and combinations with a hypothesised link to 7 flight niches. Metrics pertaining to agile flight strongly constrain shape, with hovering, diving and hawking birds developing optimal planforms. Marine soarers are suboptimal for metrics linked with low cost of transport and manoeuvrable flight. Many taxa, principally passerines, are suboptimal for all studied metrics and combinations demonstrating uneven constraint on flight performace across birds. Phylomorphospace analysis suggests planform shape is only weakly influenced by phylogeny and functional optimality correlates closely with flight styles. This suggests wing shape remains a determining factor in how birds fly.

The physical demands of becoming and staying airborne place immense pressures on avian biology, which constrain phenotypic variation[1,2]. Despite this, modern birds have adopted a diverse array of flight styles and high wing shape variability[3]. Wings, the structures underpinning flight, are therefore ideal for investigating the relationship between form and function. Wing morphology has long been associated with differing flight styles[4–13] and this relationship has been well documented from linear measurement studies[3,8,12,14], as well as studies of both 2D planform shape, principally focusing on individual avian groups[15–17] and wing shape in 3-dimentions[18,19]. These demonstrate a link between shape (both in linear measurements and in planform) and performance, with distinct wing shapes linked to specific flight modes.

Recent studies of flight dynamics[18,20] have proposed that range of motion, rather than wing shape, is the principal driver of wing diversity. Furthermore, both linear measurement and planform-based studies of shape suggest that variation in wing shape is mediated by the phylogenetic effect of shared history rather than functional pressures[12,15,17]. To address whether motion is a primary driver of variation or if function drives wing shape diversity, we sampled across seven flight styles and one-tenth of avian diversity to study the intrinsic functional properties of the wing planform.

Bristol Palaeobiology Research Group, School of Earth Sciences, University of Bristol, Bristol, United Kingdom. ✉e-mail: oz22244@bristol.ac.uk; E.Rayfield@bristol.ac.uk; Phil.Donoghue@bristol.ac.uk

Previous shape-based wing studies provide valuable groundwork, establishing performance proxies and outlining the interaction between forces that affect the wing during flight, though often on small taxon sets. Tree spanning datasets of wing form have been created, though seminal datasets of linear measurements, like that of Rayner[3], were never published.

A theoretical morphospace approach is used to produce a matrix of theoretical wing planform shape variation, from which optimal theoretical shapes for specific flight styles are identified through functional testing. By testing theoretical shapes rather than empirical wings, theoretical morphospaces allow direct investigation of performance within a global shape environment[21]. This both permits the identification of optimal forms (shapes that perform best in a global landscape for a given trait) and avoids the assumption that any empirical shape is optimal. We produced a theoretical morphospace (Fig. 1) of avian wing planform shape variation that encompasses and expands upon the range of variation in nature. Theoretical wing planform shapes were then sampled densely and evenly within this theoretical space and subjected to functional analysis. Functional performance surfaces for the theoretical morphospace were then derived. These surfaces were created for four proposed relationships between morphology and function, which act as proxies for performance, and a series of Pareto-rank-optimised metric combinations.

An originally collected dataset of 1139 extant avian planforms (36 of 41 orders) was then superimposed over this analysis of theoretical shape to determine whether extant empirical wings map to the hypothesised flight optima established through testing of theoretical morphology. The optimal peaks produced by these metrics and combinations are hypothesised to be optimising factors for flight-specific styles found within birds, allowing for direct predictions of which bird groups should have optimal planforms for which metrics (Table 1). If shape is a driver of functional diversity, we expect empirical wing planforms to map to optimal theoretical shapes for specific flight styles or combinations of functions. Alternatively, if empirical shapes are functionally suboptimal, the influence of function on wing planform diversity must not be as overwhelming as it is often perceived.

We also test for the degree to which phylogeny explains shape variation using ancestral state estimation and a phylomorphospace approach. Our results indicate that phylogeny does not explain much of avian wing shape variation, but for more biomechanically intensive flight styles, wing shape correlates closely with hypothesised functional optimality.

## Results

### Theoretical shape
The theoretical morphospace generates two main principal components of variation (Fig. 2a) that represent a combined 75% of total disparity. Over half of all disparity is represented on PC1 (53.75%), which describes the breadth of the wing chord, because all planforms are scaled to identical length. Low PC1 values represent more slender theoretical shapes, with higher values representing broader forms. PC2 (21.96% of total disparity) principally describes the tip-base weighting of the planform, with low PC2 values characterised by broader tips and high PC2 values by broader bases. PC2 also characterises primary feather emargination, which is manifest in theoretical shapes as a bifurcation of the planform tip. Primary feather splay is most pronounced in shapes with low PC2 values, though less noticeable in shapes with low PC1 values. This is because a slenderer shape causes disruption of the tip and self-intersection during the theoretical shape generation, resulting in impossible forms (grey areas in Figs. 1–5). PC3 characterises only 9.14% of variation, pertaining to the anterior-posterior position of the tip. Shapes at the extremes of theoretical space point anteriorly in a U shape or posteriorly in an N shape for low and high values of PC3 respectively (Supplementary Fig. 1). When projected into the theoretical morphospace (Fig. 2), empirical shapes occupy a large proportion of potential variation, though regions of the lower left and of the righthand quadrants remain unoccupied, with most taxa clustering around two peaks in morphological disparity (Fig. 2b).

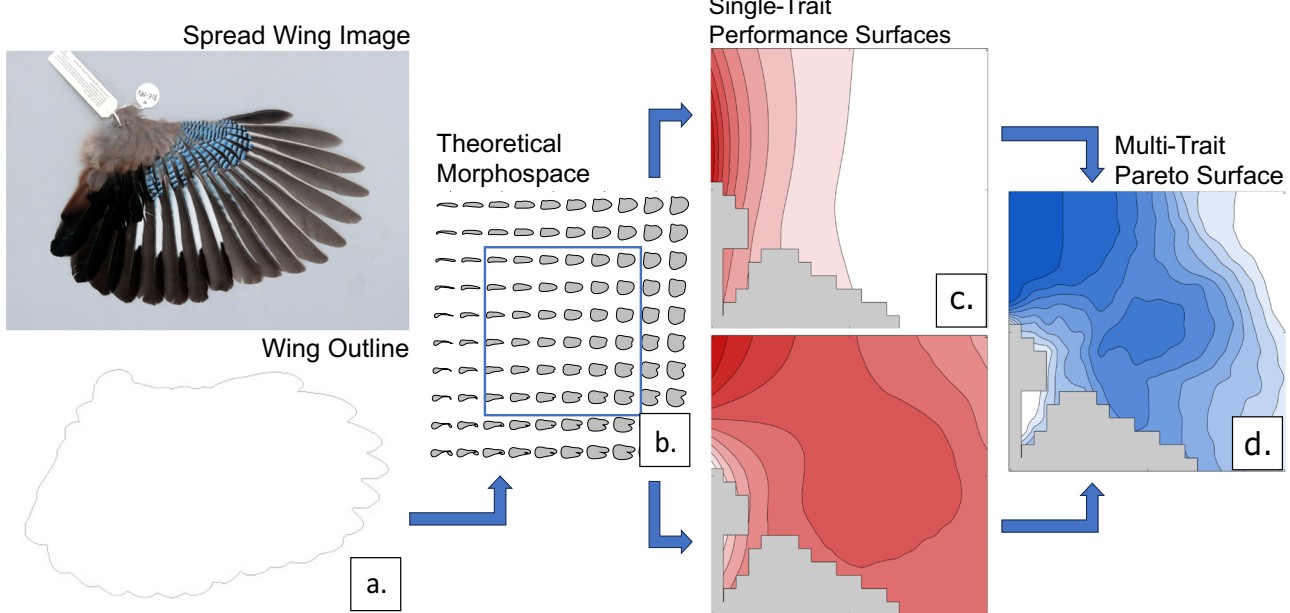

**Fig. 1 | Workflow for theoretical morphospace analysis and functional testing. a** Reduction of wing image to an outline and capture of shape information by EFA, **b** generation of theoretical wing shapes, the blue square represents the extent of space occupation by empirical shapes, expanded beyond by 20%, **c** calculation of relative performance for theoretical shapes to produce performance surfaces, and **d** Pareto ranking of multiple performance surfaces to examine trade-offs in optimality.

**Table 1 | Functional metrics and combinations analysed**

| Functional metric | Aspect ratio (AR) | Second moment of area (2MA) | Pitch agility (PA) | Tip angle (TA) |
|---|---|---|---|---|
| Derivation | Length relative to breadth of the planform, relative thickness in scaled planforms | Breadth of the planform along its length, weighting tip to base | Ability for planforms to become unstable with increased pitching angle | Inside angle of the tip of the planform, angle of the leading-edge tip in planforms with splayed primary feathers |
| Proxy | Cost of Transport | Manoeuvrability | Agility | Lift/Drag |
| Higher | Lower Transport Cost | Higher Manoeuvrability, Higher Drag | Increased Ability for Acrobatic Flight | Reduced Drag |
| Lower | Higher Transport Cost | Lower Manoeuvrability, Lower Drag | More Stable Flight | Increased Lift, Faster Take-off |
| References | 3,10,37 | [4,70] | 66 | 13 |

| Flight behaviour | Marine soaring | Aerial predation | Thermal soaring | Diving (wing-assisted) | Active hovering | Long-distance migration | Burst flapping |
|---|---|---|---|---|---|---|---|
| Primary form constraints | High efficiency, low sink rate | High force generation, tight turning | Low sink rate, large wing area | High force generation, tight turning | Rapid response | High efficiency | High force generation |
| Optical metrics | Drag reduction/ Manoeuvrability | Lift generation/ Agility/ Manoeuvrability | Drag reduction | Lift generation/ Agility/ Manoeuvrability | Agility/Manoeuvrability | Cost of transport/Drag reduction | Lift generation |
| Proxy | 2MA + TA | 2MA + PA + TA(LOW) | TA | 2MA + PA | 2MA + PA | AR + TA | TA(LOW) |
| Example taxa | Albatrosses Petrels Manx-Shearwater | Swifts Swallows Flycatchers Nightjar | Buzzards Condor Eagles Falcons Harriers Kites Osprey Vultures Cranes Storks | Penguins Dippers Auks Diving Petrels | Hummingbirds | Examples from: Geese Terns Plovers Stints Sandpipers and Others | Turkey Peafowl Curassow Pheasants Tinamou |
| References | 73,74 | 67 | 75 | 76 | | 77–79 | 3 |

(Upper) Functional metrics implemented, the flight proxies they represent and how relative values effect performance. (Lower) Flight styles analysed, along with the principal constraints imposed on that style of flight and examples of taxa included in the analysis that exhibit that form.

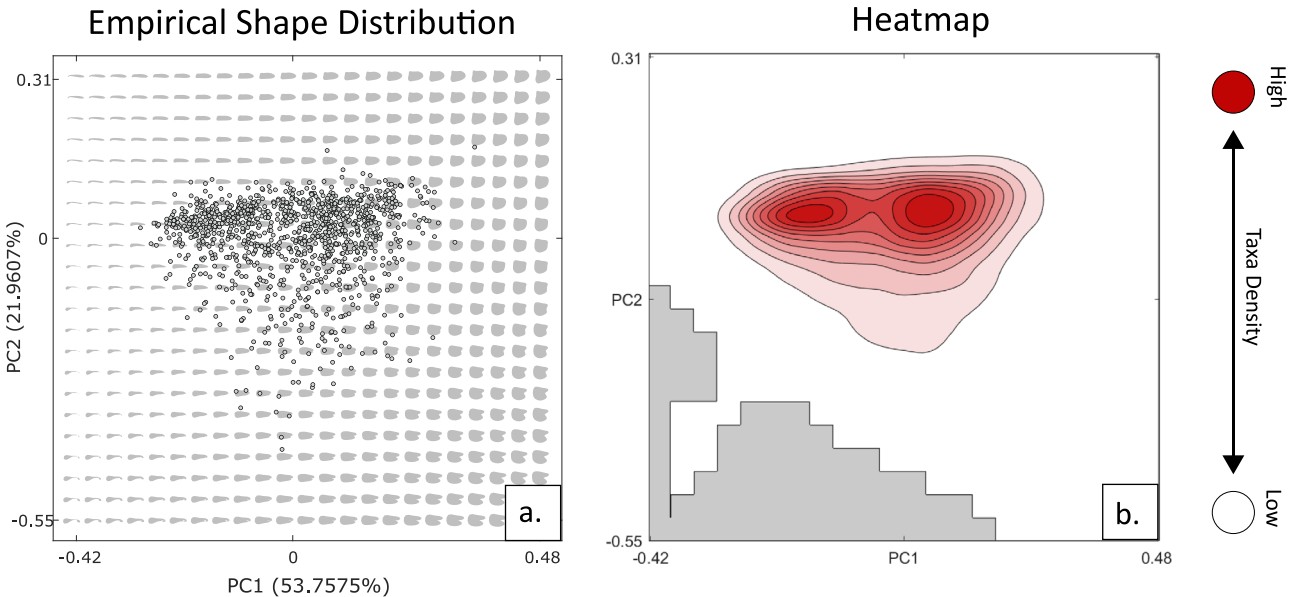

**Fig. 2 | Morphology of theoretical shapes and distribution of empirical taxa in morphospace. a** Theoretical morphospace plot illustrating the form of theoretical shapes in grey, with empirical taxa marked by block outlined grey dots. **b** Kernel density distribution of empirical taxa across the first two Principal Component axes in unexpanded morphospace, darker shading represents greater taxa density.

**Fig. 3 | Performance surfaces for the four functional metrics examined.** Darker colours represent higher optimality with empirical taxa marked as black dots. **a** aspect ratio; **b** second moment of area; **c** agility; **d** tip angle.

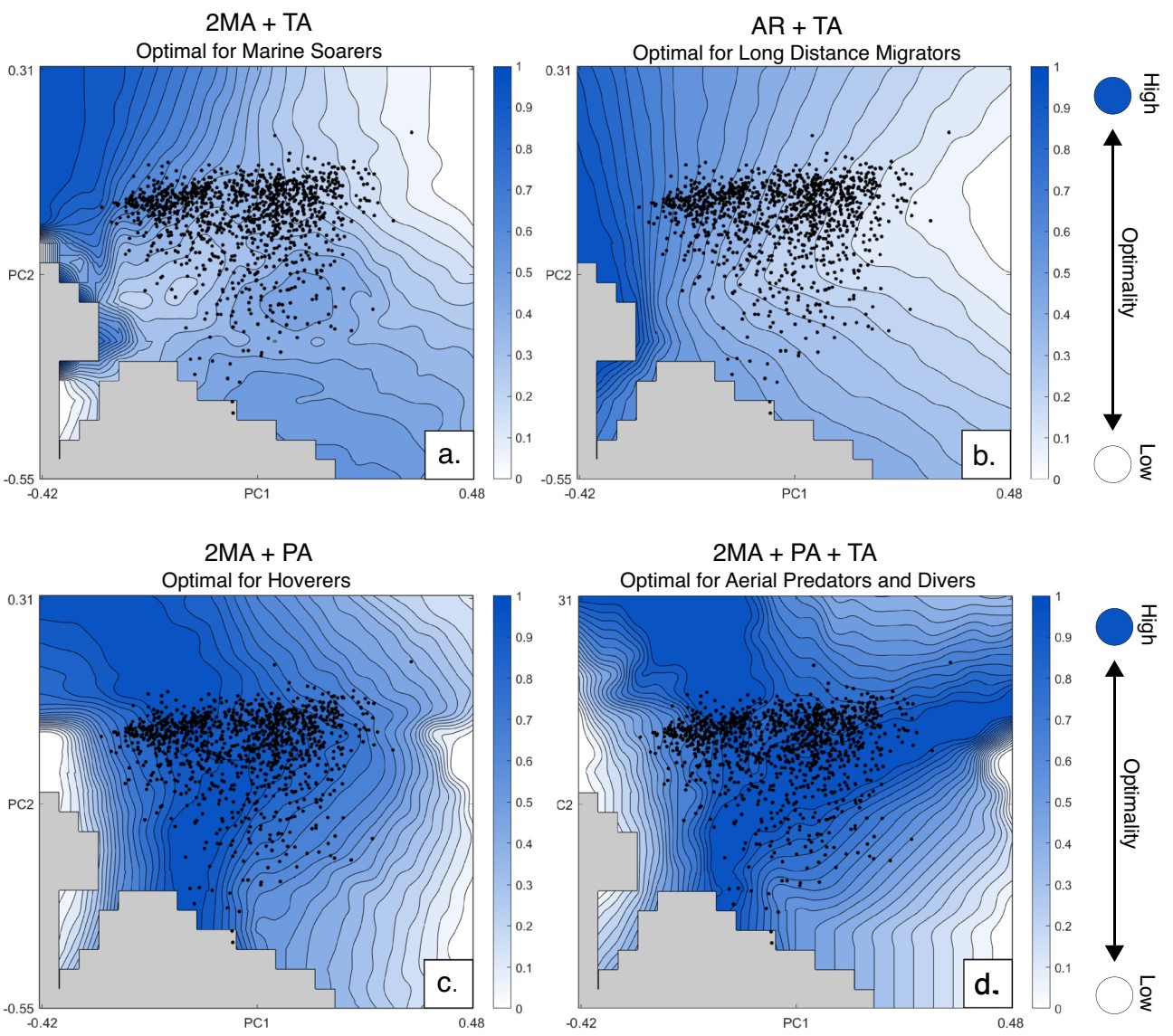

**Fig. 4 | Optimality surfaces for the four Pareto-optimal trade-offs examined.** Darker colours represent higher optimality with empirical taxa marked as black dots. **a** second moment of are and tip angle combination; **b** aspect ratio and tip angle combination; **c** second moment of area and pitch agility combination; **d** combination of second moment of area, pitch agility and tip angle.

## Hypothesised functional performance

The performance surface (Fig. 3a) for planform aspect ratio illustrates a single optimal peak with minimum values on PC1 and median values on PC2. Optimality for aspect ratio drops downslope to higher values of PC1. The area on the extreme left of the graph is characterised by very thin, highly pointed theoretical shapes, where interference from the emarginated primaries creates impossible space (grey regions, see Methods).

The performance surface for planform second moment of area exhibits two optimal peaks (Fig. 3b). The primary peak resides in space with low PC1 and high PC2 scores, characterised by slender, broad-tipped theoretical shapes. A second, locally optimal, peak exists with PC2 values at the median and PC1 values slightly above the median. This peak is characterised by broadly rectangular theoretical planforms.

The performance surface for planform pitch agility (Fig. 3c) is contoured with an oblong optimal peak projecting parallel to PC2 originating from approximately PC1 = −0.3. This peak comprises sub-ovate theoretical wing planform shapes with rounded tips and slight tapering from root to tip. Some planforms in lower PC2 optimal space

exhibit slight division in the tip but forms with more pronounced primary feather divisions fall outside of optimal space.

The optimal region for the tip angle performance surface (Fig. 3d) mirrors that of aspect ratio, characterised by long slender forms increasing in optimality with lower PC1 values. No sampled taxa occupy the optimal peak, though shapes with emarginated tip morphologies reside farther upslope than non-slotted forms.

## Functional trade off and Pareto optimality

We examine combinations of performance metrics reflecting trade-offs between antagonistic functions. Trait combinations were calculated using a Pareto ranking system[22] and Goldberg[23] ranking algorithm to optimise for trade-off between multiple traits (see Methodology). The combination of planform second moment of area and tip angle (Figs. 4a and 5a), hypothesised as optimal for marine soarers, produces a Pareto surface broadly mirroring that of the aspect ratio metric. The plot of aspect ratio and tip angle (Figs. 4b and 5f), proposed as an optimal combination for migratory taxa, also presents optimal shapes at low PC1 values. This graph most closely resembles aspect ratio, with

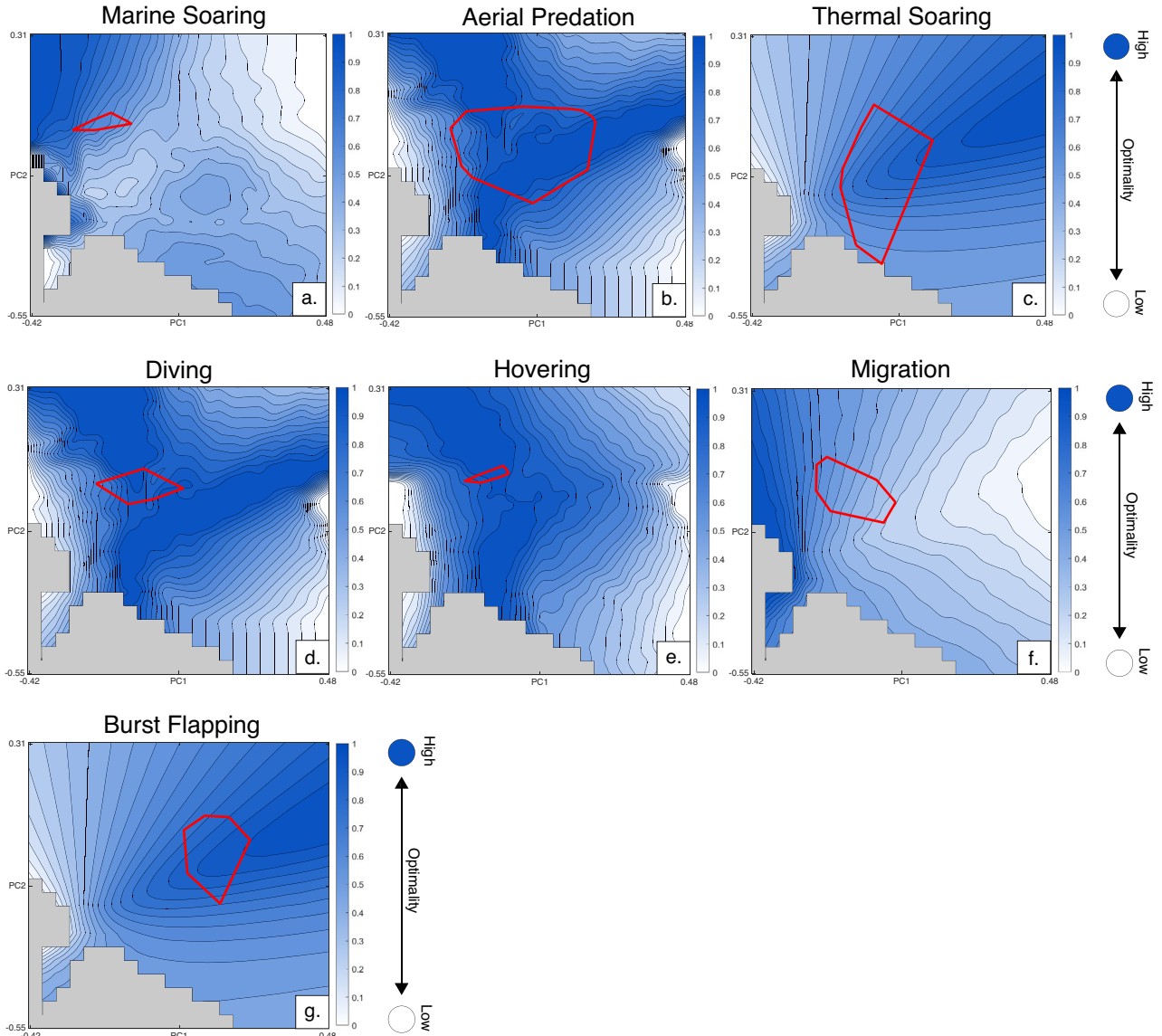

**Fig. 5 | Flight style convex hulls overlaid onto relevant performance surfaces. a** marine soaring; **b** aerial predation; **c** thermal soaring; **d** diving; **e** hovering; **f** long-distance migration; **g** burst flapping.

only a slight expansion of more optimal space downslope for the extremes of PC1.

The Pareto optimality surface for second moment of area and agility (Figs. 4c and 5d, e) serves as a proxy for acrobatic flight, such as hovering, and features a large ridge of optimal space in negative PC1 values, decreasing downslope to both the left and right of the graph. A second moment of area, agility and broad tip angle combination (Figs. 4d and 5b), which we suggest is optimal for aerial predation and wing-assisted diving, resembles that of the second moment of area:agility space, with the addition of a second branching area of optimal space for higher PC1 and median PC2 values. Several empirical planforms are optimal for this combination of traits including those with both acute and obtuse tip angles (Fig. 5b). The mean performance distance from the optimal peak for aerial predators is 0.0952, indicating over 90% similarity with optimal planform shape.

**Empirical occupation of shape space**
At order-level (Fig. 6), there is substantial overlap between the different bird lineages when projecting the empirical shapes onto the theoretical morphospace. Plotted as a heatmap (Fig. 2b), two broad

groupings emerge. Waterbirds and Apodiformes comprise the majority of the lefthand group, with low PC1 values, while the right-hand group, with median PC1 values, is dominated by members of Passeriformes as well as taxa from Cuculiformes and Galliformes. Overall, this distribution closely resembles the quadrant morphospace created by Rayner[3], derived from aspect ratio and wing loading values. Projecting phylogeny into the morphospace (Fig. 7a) further develops this evidence of homoplasy, with substantial overlap in morphology between taxa from disparate lineages. Ancestral state planforms group tightly within the same region of shape space as modern taxa. A very weak but significant phylogenetic signal was found in the shape data ($K_{mult} = 0.1751$, $p = 0.001$) using a multivariate K statistic[24]. Two additional tests of phylogenetic signal were conducted using the taxa sets employed by Wang and Clarke[15] and Baumgart et al.[17] (Supplementary Data 3). These were performed using the same methods employed in the above analysis with the aim of testing whether the method used is consistent across datasets. Reproduction of the Wang and Clarke dataset produced a large phylogenetic signal ($K_{mult} = 0.9293$, $p = 0.001$) while reproduction of the Baumgart et al. dataset produced a signal of $K_{mult} = 0.088$ ($p = 0.002$). When projected into

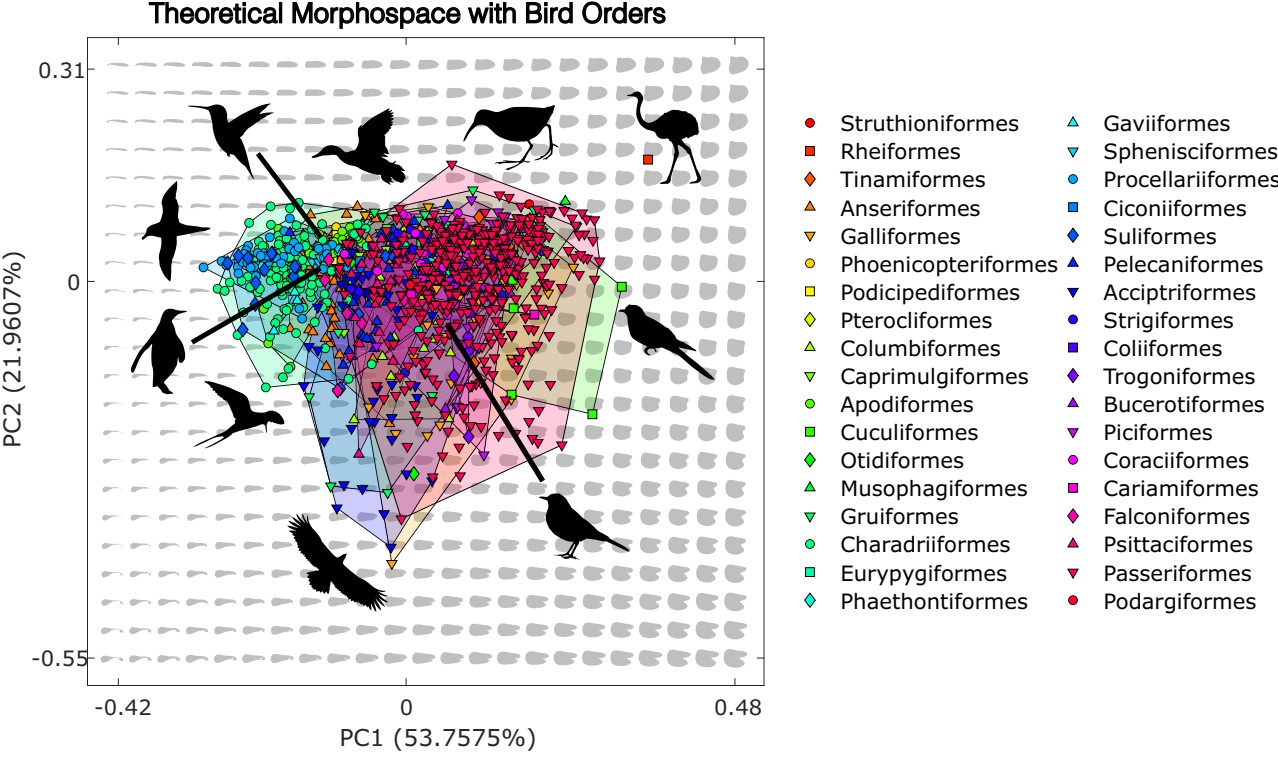

**Fig. 6 | Theoretical morphospace with taxa separated by order.** Coloured dots represent positioning of empirical wing shape with approximate location of selected bird groups. Example silhouettes were obtained from PhyloPic (http://phylopic.org).

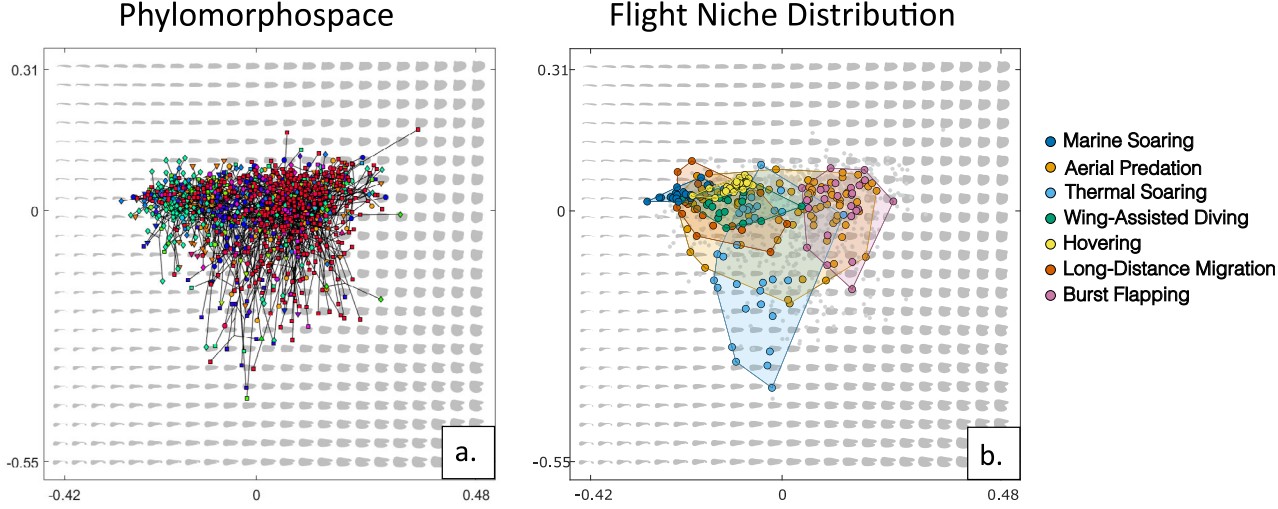

**Fig. 7 | Divisions of morphospace by phylogeny and behaviour. a** Phylomorphospace illustrating positioning of ancestral states and clustering of morphospace.
**b** Theoretical morphospace with convex hulls illustrating locations of example taxa for specific flight styles.

morphospace (Fig. 7b), flight style groups exhibit considerable overlap, though some patterns are visible. Most dynamic soaring, hovering, diving and long-distance migrating birds group with the lefthand waterbird-dominated cluster, while burst-flapping poor flyers plot mostly on the right, overlapping considerably with the passeriforms.

## Discussion

### Empirical occupation of theoretical morphospace
Uneven occupation of theoretically available planform shape space by the empirical taxa may result from several factors unrelated to function. The simplest is that of time constraint, although the radiation of modern birds has been explosive since the Cretaceous[25], providing

ample opportunity for evolutionary exploration of shape space. Unoccupied theoretical space could also be explained through the characterisation of emarginated primary feathers. This may influence the position of taxa, though it is unlikely to account for the large regions of unoccupied space. While distal primary spread is linked to tip shape[14,26,27], the distribution of groups in morphospace greatly resembles previous studies[3], which do not consider wingtip shape. This suggests that broader trends in planform shape are preserved across our dataset. Furthermore, the emarginated morphology is only present in theoretical shapes with high PC1 values, while a large area of low PC1 space remain unoccupied. Some of this region is covered by impossible space (self-intersecting planform shapes); however, this

does not account for most of the unoccupied shape space. There exist possible theoretical morphologies with slender tips and robust bases that are not manifest in any sampled extant bird group. This suggests that taxon clustering and the large areas of unoccupied space do not result from issues of dataset sampling.

Additional constraints on planform shape may come from the relationship of the wing as part of a larger organism. In taking this system as a whole, there may be multiple sources of shape-constraint on the planform imposed by the biology of the rest of the animal, as is the case with many-to-one mapping observed in other animal groups[28]. Alternatively, the use of wings for multiple flight and additional non-flight purposes, such as signalling, may result in pleiotropy, which constrains the evolvability of optimal forms[29,30]. This would result in the wings of taxa which employ their wings for behaviours like sexual selection being more constrained away from flight optimality than those of taxa which do not undertake these activities.

## Phylogeny

The weak phylogenetic signal observed implies that a phylogenetic effect of planform shape cannot be discounted but the effect on overall morphological variation is likely minimal, providing additional evidence against time constraint. Occupation of limited shape space by diverse bird lineages, coupled with weak phylogenetic signal, suggests that morphological convergence within birds is mediated by factors other than phylogeny. This contradicts previous studies of avian wing variability, most notably Wang and Clarke[15] who, in their analysis of wing planform shape, propose a strong phylogenetic signal. Discrepancy is unlikely to result from differences in the method of computing phylogenetic signal as our phylogenetic signal tests of reconstructed datasets for both the Wang and Clarke and Baumgart et al.[15,17] produce results consistent with the original analyses. Instead, the result may result from differences in the size and composition of the studied datasets, as these differ by an order of magnitude. Additionally, differences in dataset sampling protocol may play a role, particularly the inclusion of covert extent by Wang and Clarke[15]. With dataset-specific and phylogenetic factors accounted for, what remains is the suggestion that the convergence observed in the dataset is the result of functional constraint.

## Low cost of transport flight

No empirical planform shape comes close to occupying the optimal peak for planform aspect ratio, with the closest shapes plotting only halfway up the optimality slope (black dots, Fig. 3a). The group with the highest aspect planforms, marine soarers, form the closest position to the optimal peak, though their positioning is still suboptimal (Fig. 5a). While theoretical morphospaces are Euclidean for any given set of principle components, the difference in performance between two equidistant pairs of points varies depending on the topology of the performance surface. Since performance values are calculated directly from the even grid of theoretical planforms, with one optimal planform shape, the difference in performance value therefore gives the distance between any platform or mean of a group of planforms from that optimum across the surface. The remainder of the distance from the optimum (a distance of 0) therefore represents an average percentage of shape similarity. For aspect ratio, the mean difference in performance from the peak value for marine soarers is 0.3455, representing an average similarity of 65% to the optimal shape for aspect ratio. Most other realised wing planforms occupy lower aspect ratio regions, hypothesised to be non-optimal space. Two empirical shapes with high PC2 values overlap with impossible space, a consequence of flattening higher-dimensional space into a 2D grid for visualisation.

The lack of occupation of the aspect ratio optimality peak by bird groups hypothesised to be optimal for low cost of transport flight (Table 1), suggests that, while reducing transport cost is essential for migratory and marine soaring animals, this alone is not enough to completely constrain planform shape. Even the highest aspect planform in the dataset, that of the Southern Royal Albatross (*Diomedea epomophora*), remains in non-optimal space. This may be because marine soarers are constrained by the need to reduce drag and adjust to changing wind currents. Dynamic soaring requires frequent turning adjustment to maximise wind wave interaction[31], thus additionally requiring high manoeuvrability as well as acute tip angles to reduce tip vortices and improve performance[32,33]. Testing this combination of functional constraints, marine soaring planforms remain suboptimal, with a mean peak distance of 0.3952 (Fig. 4a). Giant extinct pelagornithids possessed wings with lengths at the limits of what is considered biologically viable[34]. With a proposed conservative aspect ratio of 13[35], *Pelagornis sandersi* would plot just off the optimal peak (aspect ratio of 13.8), suggesting that optimal shapes for marine soaring, while theoretically possible, lie at the very limit of avian biology. As some of the longest-winged extant birds, albatrosses demonstrate another potential reason for the lack of development of longer, thinner planforms: the difficulties of take-off and landing without the assistance of wind[36]. Since landing to nest is a requisite for species survival, breeding presents a clear constraint on the development of more extreme shapes, which hinder take-off and landing. These results concur with the interpretations of Rader et al.[19] who identify the necessity of high lift production as a reason the discrepancy in lift-to-drag coefficient between gliding and non-gliding birds. The pattern of near optimality continues for the planforms of long-distance migratory taxa demonstrating that, while minimising cost of transport and induced drag are proposed to be essential for cross-oceanic migration[3,4,37], additional factors constrain planform shape. Proximity of some migratory taxa to the optimal peak for the combination of aspect ratio and tip angle does, however, suggest some degree of constraint.

The development of emarginated tips presents an alternative method for reducing drag and improving flight performance[38,39], as each individual primary feather acts in miniature as its own wing, producing an individual wake. This morphology, found in thermal soarers such as condors[38], allows birds to simulate much longer, high-aspect planforms while avoiding the drawbacks. We attempt to account for this effect by only considering the leading-edge point in the tip angle calculation, though this still likely underrepresents the drag-reducing effect of a slotted planform tip. Resultingly, thermal soarers appear largely unoptimised in this analysis (Fig. 3d).

## Acrobatic flight

To contrast flight niches focussed on reducing drag and conserving energy, planforms of taxa which occupy the acrobatic niche of aerial predation appear highly optimal. In the pitch agility landscape (Fig. 3c), the optimal performance peak is densely occupied including the most densely populated region of morphospace. Taxa occupying the peak include multiple bird orders and flight niches, including Podicepediformes (grebes), Phoenicopteriformes (flamingos) as well as taxa hypothesised to be optimised for this flight style, such as Apodiformes (swifts, tree-swifts and hummingbirds) (Table 1). The dense occupation of this peak suggests that the ability to switch to unstable flight to initiate manoeuvres is a fundamental constraint in many modern birds. The most optimal theoretical planforms are pointed, running contrary to previous analyses, which show that tip shape is unrelated to an ability to navigate aerial obstacles[13]. The expansion and denser occupation of the optimal peak for acrobatic flight to include many aerial hawking species demonstrates that species occupying this niche have been strongly optimised by functional constraint.

## Hovering and diving

Taxa hypothesised to be optimal for hovering flight specifically, occupy the region of theoretically optimal space in both the planform agility performance surface (with a mean variation of 0.1509 and

0.1455 from the peak) and the planform second moment of area:agility Pareto space (mean variation of 0.0287 and 0.0636) (Fig. 4c). This adds further credibility to the suggestion that more extreme flight niches experience tighter functional constraint. Penguins (Sphenisciformes) also occupy the peak, though with greater spread in optimality. The mean peak distance for penguin wings is 0.1623 (83% similarity) and some taxa posessing wings with upwards of 97% similarity despite water being a much denser medium than air. Penguins fit neatly within the class of wing-assisted divers and possess planforms specially adapted for agile movement in water[40]. The constraints of medium density between water and air seem to have little effect on the optimal shape of planforms for agility. The planforms of hummingbirds also appear similarly constrained and converge on an optimal shape, despite these taxa utilising a flight pattern more akin to insects than the rest of avians[41]. A large portion of thermal soarers occupy the adaptive peak for this functional combination. There is a strong link suggesting the benefit of emarginated primary feathers in agile flight[42], though this particular advantage remains untested by this analysis.

### Other flight modes

Unlike the other performance metrics tested, the constraint imposed by planform second moment of area (Fig. 3b) appears to function more as a threshold with most taxa occupying a high, though suboptimal, plateau. This plateau encompasses the densest population of passerines, as well as taxa from multiple other orders, suggesting that manoeuvrability partially constrains all volant planforms but does not drive optimality. The flightless Rhea is positioned farthest downslope from the plateau in the least optimal position, further supporting this. Once surpassed, diversification is mediated by other factors[12]. The broadest tipped, burst flapping planforms reside within and adjacent the optimal peak for tip angle (inverted so broad tips are optimal), supporting the hypothesis that the wings of these taxa are optimally designed for quickly producing lift (Fig. 5g). This may suggest that for some primarily ground-dwelling taxa, this function greatly constrains planform shape, though the majority of burst flappers reside downslope.

None of the above evidence supports the existence of an ideal generalist morphology, where optimal taxa would occupy the pareto-optimal peak for the combination of all four metrics. Instead, the area of highest performance for this metric combination encompasses the greatest number and variety of taxa of any optimal peak, including specialised taxa like albatrosses.

The apparent lack of optimal planforms within some groups may result from the use of planforms in analysis. While the fully spread planform represents a shape realised by the wing for only a small fraction of a flap cycle, utilising planforms normalises wing shape in a manner that permits large-scale comparative analyses. It is possible that flight in less optimally performing birds relies less on the fully outstretched wing. Modelling wing shapes in multiple semi-closed configurations may account for this optimality discrepancy, or factors alternative to shape may be responsible. Additionally, functional constraint may be relaxed in smaller birds due to the decrease of muscle power with larger body masses. Smaller birds have proportionally stronger flapping ability, which may help ease the aerodynamic constraints experienced by larger birds, potentially explaining the relative lack of optimal planforms within Passeriformes. This is not a universal trend, however, as hummingbirds possess optimal planforms and show functional constraint despite their diminutive size.

Analysing function directly has demonstrated that for bird groups, particularly those with agile or intensive flight strategies, the shape of the wing planform does act as a constraint on function. This appears to run counter to the observations suggested by Baliga et al. and others[18,20] who posit that shape has no primary impact on variation. However, the constraint on shape is not recognised for all birds. Most birds sampled exhibit little shape-based constraint on function, and for some functional metrics, constraint exists to a threshold, as is

the case with manoeuvrability. It stands to reason that, for less specialised fliers such as passerines and landbirds, factors such as range of motion play a more primary role in avian flight diversity. This result also partially contradicts the conclusions of Wang and Clark[15] who suggest only a tenuous relationship between variation and function. This may be the result of our increased dataset size, a possibility supported by the similarities between these results and those of Rayner[3], whose dataset contains over 1500 species.

By analysing function of possible rather than empirical shapes, theoretical morphospace permits examination of wing shape distinct from the adaptationist assumption of functional optimality. Through subsequent comparison of optimal theoretical planforms with the variation found in birds it is possible to assess the actual optimality of avian wings and how well birds meet flight performance criteria relative to idealised forms. Many, but not all, bird wing planforms appear largely unoptimised for the constraints of flight, and in contrast to previous studies, there appears to be little phylogenetic effect on planform shape. Even the best performing birds are still far from optimal for individual performance metrics and manoeuvrability acts as a floor rather than an adaptive performance peak. Instead, agility appears to be a primary force constraining avian wing planforms, with multiple flight niches and bird orders converging on an optimal agile planform. Occupation of theoretical optimal peaks by hoverers, divers and aerial hawkers, and the proximity to optimal peaks of marine soarers and migratory taxa, suggests that more extreme flight niches experience higher functional pressure towards optimal forms than others. This work demonstrates that, for at least some bird groups, shape remains a primary driver of functional variation while the identification of a functional threshold for manoeuvrability demonstrates that it is not the sole factor that constrains the function of bird wings.

## Methods
### Data collection

All data collected for this work is freely available for download and use without permissions. 1139 images of bird wings spread to their maximum extent were sourced digitally and captured directly from museum ornithology collections, principally from the Burke Museum of Natural History and Culture (UWBM) Seattle, United States, the Puget Sound Museum of Natural History (PSM) online Wing and Tail Collection and the Natural History Museum at Tring (NHMUK), UK. Additional wing images were sourced from the online collections of:

- The Auckland War Memorial Museum (AK), Auckland, New Zealand
- The Field Museum of Natural History (FMNH), Chicago, United States
- The Musée des Civilisations de Cote d'Ivoire (MCCI), Abidjan, Ivory Coast
- The Museum of New Zealand Te Papa Tongarewa (NMNZ), Wellington, New Zealand
- The Slovenian Museum of Natural History (PMSL), Ljubljana, Slovenia
- The Naturalis Biodiversity Center (RMNH), Leiden, Netherlands
- The Smithsonian National Museum of Natural History (USNM), Washington DC, United States

as well as from online wing and feather resource collections and the published literature. A complete taxa list is provided in Supplementary Information (Data 1). For data collected directly from museum specimens, individual wings were photographed from above on a plain background. Righthand wings were selected where possible, and where needed, lefthand wings were mirrored to ensure data consistency. This resulted in a planform image, the external edge of which was then defined using the outline selection tool in Fiji (version 1.54)[43] (Supplementary Data 4). Henceforth, planform refers to the outline of

the outermost silhouette of the wing. 2D planform wings, characterised as outline shapes, were chosen due to the online availability of images and high degree of variability in the preservation of 3D wing reference collections. Furthermore, previous large-dataset studies of wing shape and flight performance have employed 2D shape, demonstrating its efficacy for disparity analysis[15]. 2D shape can be easily defined using Elliptical Fourier Analysis (EFA)[44,45], permitting accurate silhouette generation. EFA was completed in MATLAB following the protocol established by Deakin et al.[22]. In addition to the wing planforms, the unpublished dataset produced by Rayner[3] has been reconstructed, with its original measurements (Supplementary Data 5).

The data was split at order level and includes 36 of 41 extant bird orders, with additional subsets created to characterise six niches derived from Rayner[3] and Taylor and Thomas[12]: marine soaring, aerial predation, thermal soaring, wing-assisted diving, long-distance migration, and burst flapping. An additional class of hoverers was added to encompass hummingbirds. The taxa in these convex hulls do not represent a comprehensive list of all taxa with a particular flight niche in the dataset, but were chosen from the literature to encompass the approximate maximum area of occupied morphospace in which animals employing this niche may reside. This research considered flight niches individually, though many taxa may be optimised for a combination of multiple niches.

For phylomorphospace analysis, a strict consensus time-scaled supertree was generated from the all-taxa supertree on birdtree.org[46,47] using the Hackett[48] backbone. Consensus was achieved using the consensus edges tool in phytools[49] with the least squares method (Supplementary Data 2). Two taxa, the Mexican Hermit Hummingbird (*Phaethornis mexicanus*) and Wilson's Phalarope (*Phalaropus tricolor*), were excluded from phylomorphospace analysis as they are absent from the birdtree.org taxa list. Phylogenetic signal was assessed using the physignal command in the R package geomorph, which returns a multivariate K statistic (kmult) of the relationship between phylogeny and shape. Ancestral shapes were then computed using the command gm.prcomp for maximum likelihood ancestral state reconstruction[50,51]. This method was chosen due to the applicability of these functions for working with coordinate data, matching with the outputs of EFA. All R analyses were performed in R version 4.2.2 and Rstudio version 2022.12.0[52,53]. The code and algorithms for computing performance characters were written in MATLAB version R2022B by Deakin[22] and modified to work with the constraints of planform shape. The code and all datasets are available in Supplementary Information Code 1.

## Theoretical morphospace

The morphospace constructed here follows the approach utilised by Raup[54] and outlined in previous studies[22,55–62], producing theoretical shapes using constructor variables derived from measurements of empirical taxa. While some suggest that this method does not yield truly theoretical shapes[21], it remains the prevailing method of constructing theoretical space. Using Fourier Shape Analysis via the method established in Deakin et al.[22], we characterised each planform with 1200 outline landmarks and 5 shape harmonics. This number of landmarks is well beyond the threshold of convergence and was the maximum number possible given the resolution of each outline. Plotting the formula for the elliptical harmonics produced a 2D representation of empirical shapes, the morphological disparity of which was characterised through Principal Components Analysis (PCA). Since altering formula parameters for shape proportionally changes the resulting outline, changes in the formula were used to generate an even grid of theoretical shapes encompassing the entire range of shape variation in the empirical taxa[44]. This theoretical shape grid was then expanded 20%[22] resulting in 23 ×22 grid of 506 theoretical wing planform shapes plotted evenly across the principal components space (Fig. 1). Expansion of the theoretical grid ensures that a larger area of theoretical morphology is functionally tested than exists in nature, allowing for the potential identification of optimal morphologies that are not realised in living animals. Expansion of this theoretical morphology space was constrained at 120% of observed empirical variation focus on possible (i.e. non self-intersecting) shape, and forms which resemble the planforms of wings, as well as the practical limitations of computational analysis. The low number of shape harmonics used allowed for maximisation of possible landscape space, as higher harmonic levels introduced greater regions of space with self-intersecting forms due to overlap from generation of forms with emarginated primary feathers. Self-intersecting shapes are theoretically impossible for 3D objects presented two dimensionally[63,64], so 75 self-intersecting forms were removed from functional analysis, represented by the grey region in the performance space plots. All theoretical planforms are identically size scaled by standardising the parameter of the first elliptic to isolate the shape element of variation and remove any length-based effects which would obscure analysis of a dataset with a high degree of size variation.

Extrapolating theoretical planform shapes with emarginated primaries results in a split point morphology in the theoretical shapes, which is unlike the splay of primary feathers found in birds. Since the distal-most primary feathers are decoupled from the ligaments that spread the rest of the wing primaries[65], this morphology may represent underlying trends in the placement of feathers during conservation, but this is unlikely as the taxa sampled derive from multiple independent museum collections and are of varying age. It is more likely an emergent feature of characterising emargination by EFA, as it is preserved at higher harmonic values.

## Performance proxies

From previous flight studies[3,4,12,13,66] we selected four functional metrics to test planform functional optimality (Aspect Ratio (AR), Second Moment of Area (2MA), Pitch Agility (PA), and Tip Angle (TA)). These metrics incorporate traditional analytical elements of flight such as lift and drag, while providing a more direct link to specific flight behaviours (Table 1). While body size is a critical element of flying animals and is linked to flight ability through metrics like wing loading[3], relative size is not computable for theoretical shapes. As a result, only functional metrics divorced from body size were included. We then linked a single metric or combination of these metrics to wing planforms associated with a given flight style niche (Table 1b), hypothesising that birds with distinct flight styles would optimise for these performance traits. For example, aspect ratio is closely linked with cost of transport, with higher-aspect (longer and thinner) planforms reducing the energy cost of flight[3,12]. For this reason, high aspect ratio is considered optimal, particularly for birds with soaring and long-distance migratory niches such as albatrosses and other procellariiforms, where energy conservation is essential. Second moment of area describes the distribution of the aerodynamic surface relative to the root of the planform, a factor which correlates with generation of rotational moment[4]. This serves as a proxy for manoeuvrability, defined by Norberg and Rayner as the ability to turn with a tight radius around the yaw axis[3,8]. We therefore hypothesise high second moment of area to be optimal for multiple flight styles including marine soaring and aerial predation, and part of the combination of metrics which drive optimisation in bird groups such as swifts, hummingbirds and procellariiforms. The relationship between the breadth of the wingtip and the trade-off between lift production and drag reduction is well established[13,32,33,67]. Planforms with broader rounder tips produce greater lift and conversely more pointed tip forms produce less lift but benefit by correspondingly reducing induced drag. For these reasons, we consider broad tip angles to be optimal for birds which require high lift for short periods, such as burst flappers and hypothesise that optimally shaped wings for this metric should be found within Galliformes and other

predominantly terrestrial taxa (Table 1b). Pointed tip morphologies, conversely, are optimal for birds with flight niches prioritise energy saving, such as migrants[14,68] and acute tip angle is hypothesised to be an optimising metric for members of Accipitriformes. Agility refers to the ability to rapidly instigate change in direction, particularly at high speeds. This is linked to the ability of wings to transition from stable to unstable flight modes through change of the pitch of the wing[66,69]. We consider planforms which can more easily achieve instability as more optimal, particularly for niches that involve acrobatic flying such as aerial hawking and hovering. Thus, agility is hypothesised to be an optimising factor on the planform shapes of taxa including hummingbirds and other apodiforms (Table 1b).

### Performance surfaces and Pareto ranking

Aspect Ratio was calculated as the square of span divided by area using the formula[70]:

$$AR = \frac{R^2}{S} \tag{1}$$

Where R is the span length of each theoretical shape and S is the corresponding shape area.

Second moment of area was used as a proxy for manoeuvrability, as the ability to fly at low speeds and perform tight turns is correlated with large second moments of area[4,70]. Thus, high second moment of area is considered an optimising characteristic. For each theoretical shape, second moment of area was calculated following the method in Liu et al.[55] using the formula[71]:

$$r_2 = \frac{1}{12} \sum_{i=1}^{n} (x_i y_{i+1} - x_{i+1} y_i)(x_i^2 + x_i x_{i+1} + x_{i+1}^2) \tag{2}$$

Under this formula, each theoretical planform is represented as a polygon with an equal number of sides as landmarks i, and where x and y represent coordinates of the *i*-th landmark. This was further non-dimensionalised to remove the impact of size, producing the non-dimensionalised second moment of area $r^{\wedge}_2$.[70]

$$r^{\wedge}_2 = \sqrt{r_2/SR^2} \tag{3}$$

Agility is inversely related to stability[66,69]. High agility is critical where sudden flight changes are necessary in high-speed flight[3,69]. In many birds, these sudden changes have been linked to the ability to temporarily switch from stable to unstable flight. Harvey[66] established a metric for pitch agility which relates to the ability for shapes to become unstable at higher angles of attack. This instability is key for birds to accomplish complex acrobatic feats[66]. This pitch agility metric was used in this analysis, expressed as:

$$\frac{\Delta \dot{q}}{\Delta a} \propto \frac{\left[\left(\left(\frac{\widetilde{x_c}}{c_{r_{max}}}\right)^{0.8} c_{r_{max}}\right) - x_{CG}\right](m^{0.12})^2 S_{max}}{I_{yy}} \tag{4}$$

Where $\Delta \dot{q}$ represents the rate of change of angular acceleration relative to the y-axis[66]. Since all theoretical planforms were tested equally at a single angle, $\Delta \alpha$ remained static, allowing for output of relative values that could be plotted onto morphospace. Since all theoretical shapes are scaled, the maximum root chord ($c_{r_{max}}$) and the quarter chord of the mean chord ($\widetilde{x}_{\frac{c}{4}}$) were calculated by dividing each planform into a uniform number of equidistant strips. Maximum planform area ($S_{max}$) was calculated from centroid area and, as the shapes tested are theoretical, mass (m) was held constant. For centre of gravity ($x_{CG}$) the farthest left point on the x axis was substituted for

the position of the femoral head. As the shapes tested are theoretical rather than measurable shapes, both mass and area are reduced to standardised values. The point of rotation for each theoretical shape was chosen as the minimum x coordinate in line with the calculation for rotational efficiency. A dimensionalised variant of the second moment of inertia ($I_{yy}$) was utilised.

The angle of the planform tip and its relative pointedness is directly related to the generation of tip vortices and thus the production or reduction of induced drag[13]. Pointed tips minimise weight and reduce drag in flight, while broader tips produce greater thrust and lift at the expense of increased induced drag[13,32,33,67]. A more pointed tip is considered optimal for long duration flight behaviours, while broad tips are more optimal for behaviours involving quick bursts of speed.

Tip angle was calculated from the internal angle of the tip of each planform shape, using the rightmost point as the tip and calculating the angle from it 500 points away from the tip in either direction. For shapes with a bifurcated tip, the leading-edge point was chosen.

The relative values of these characteristics were then plotted onto the two-dimensional morphospace to produce a three-dimensional performance surface (following Deakin et al.[22]; Fig. 1). High points on this performance surface are considered to represent more optimal theoretical shapes for the given performance metric.

No individual design is optimal for a single trait and thus all biological forms are functionally bound by a combination of multiple functional characters as well as external influences[3,12]. To examine how different competing trait-based morphological drivers interact, a method of combining functional metrics by ranking was implemented, following the method developed by Deakin et al.[22]. This method is developed from Pareto ranking algorithms[23,72] and has been demonstrated as an effective means of analysing optimality with morphospaces[22,55]. Pareto optimisation identifies optimal subsets of data for multiple interacting factors where the scaling between variables is uneven and there is no given ranking for the relevance of a given performance metric[22]. This allows for ease of comparison between large datasets where taxa are differentially preferred for a given combination. To develop a landscape from the binary Pareto grouping, we instituted an iterative Goldberg[23] ranking algorithm which produced successive Pareto fronts and further developed into a Pareto Rank Ratio (PRR), reducing the bias introduced from the occupation density of performance space. The resulting rank ratio measures proportional optimality of a solution given optimal and sub optimal ranking[22]:

$$PRR = \begin{cases} 1, & R_o = 0 \\ \frac{R_s}{R_o + R_s}, & R_o > 0 \end{cases} \tag{5}$$

Where $R_O$ represents optimal rank under Goldberg and $R_S$ the suboptimal rank. Where both values equal 0, the PRR is 1. The result is a Pareto rank from 0 to 1 for each theoretical shape, which can then be mapped as a single performance surface (Fig. 1).

### Reporting summary

Further information on research design is available in the Nature Portfolio Reporting Summary linked to this article.

## Data availability

All data used in and generated by this study have been deposited and are accessible on Figshare at https://doi.org/10.6084/m9.figshare.31353850. Additionally, the previously unpublished Rayner dataset of birds has been made available as Supplementary Data 5. All data uploaded are freely available without restrictions. The original location and institution of all wing data collected for this study are available in the 'Taxa Set' file, located in Supplementary Data 1.

## Code availability

The base code used in this study both for analysis and for the generation of all figure elements is available on Github, accessible at Deakin, W. J., Rayfield, E. J., & Donoghue, P. C. theofun (Version 0.0.1) [Computer software]. https://github.com/Bristol-Palaeobiology/theofun). All executable files and modifications of the base code have been made available in Supplementary Code 1, accessible on Figshare at https://doi.org/10.6084/m9.figshare.31353850. All code files uploaded are freely available without restrictions.

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

## Acknowledgements
The authors would like to thank Jeremy Rayner for providing access to his original bird data and taxa set; Will Deakin for guidance and support in the application of the theofun pipeline; Mark Adams at the Natural History Museum, Tring, and Chris Wood and Kevin Epperly at the Burke Museum of Natural History and Culture for allowing access to their collections and facilitating data collection. We also thank artists Andy Wilson, Alexandre Vong, Ferran Sayol, and Sharon Wegner-Larsen for making available their bird silhouettes on Phylopic and the editor and reviewers of this article for their insightful feedback. The authors would also like to acknowledge the following funding sources: the John Templeton Foundation (JTF62574 E.J.R. and P.C.J.D.), the Leverhulme Trust (RF-2022-167 P.C.J.D.), the Biotechnology and Biological Sciences Research Council (BB/W00867X/1 E.J.R.). The opinions within this article are those of the authors and do not necessarily reflect those of the John Templeton Foundation.

## Author contributions
Conceptualisation: B.W., E.J.R., P.C.J.D. Methodology: B.W., Y.L., E.J.R., P.C.J.D. Software: Y.L., E.J.R., P.C.J.D. Investigation: B.W. Visualisation: B.W. Supervision: E.J.R., P.C.J.D. Writing – original draft: B.W. Writing – review and editing: B.W., Y.L., E.J.R., P.C.J.D.

## Competing interests
The authors declare no competing interests.
