## [Transparent Peer Review file · Nature Communications]

Theoretical Morphospace Reveals Mixed Optimization of the Avian Wing Planform for Flight Style.

Corresponding Author: Mr Benton Walters

Version 0:

Reviewer comments:

Reviewer #1

(Remarks to the Author)

The authors explore the range of existing avian wing shape relative to theoretical possibilities, a very fun question for the functional morphology community, and not often done. This is an important framework to think about how things have evolved the way they do and what may be guiding or constraining their direction of evolution. I made some comments below which I think could improve the manuscript, but I think it is going to be a good addition to the journal after revisions.

Line 80 – would this be better described as the wing aspect ratio? Based on the description in the next sentence, it sounds like PC1 is breadth of the wing chord relative to the wing length, not breadth of wing chord in isolation. Or is this a raw measurement of wing chord?

Line 138 – no suppl. fig number given

Lines 144-146 – dominant by passerines or by non-waterbirds? I see passeriforms, but also cuculiforms, and galliforms on the far right. With all the other clades in that region that are hidden due to density of data points, I'd recommend being more inclusive with taxa listed in this contrasting statement, especially if non-passerine passeriforms are included in the analysis.

Line 156 – extra “/” included

Line 157 – “Wang” instead of “Wange”

Line 162-163 – “burst flapping” should be hyphenated when used as an adjective

Line 164 – passerines or passeriforms? Taxonomic categories here are of the order level, so it may make more sense to refer to the group as passeriforms instead of passerines, which is a subset of Passeriformes.

Thoughts on section at 166 – What about the idea of many-to-one mapping? (See Wainwright et al. 2005: <https://academic.oup.com/icb/article/45/2/256/778330>) Or the related idea of “this shape is sufficient for the task; taking the whole organism and its collection of needs and behaviors, it’s good enough.” Wings are not in isolation of the rest of the body, which would be a source for further constraints on empirical wing shape that could limit the amount of theoretical wing shape space that is achieved in reality.

Thoughts on sections at 185 – This is very interesting. Might differences in data collection protocols affect the differences found between the authors’ phylogenetic signal and that of Wang and Clarke 2015? They also captured the covert outline too. But it’s true that scale of analysis would also play a large effect.

Line 217 (and throughout discussion) – Thinking about terms like “cost-efficient” and “optimal/non-optimal,” clear biological context matters with these sorts of statements. An albatross wing has a high aspect ratio and therefore optimal for soaring in an open marine environment, but it is not optimal for maneuvering in a forest. Relative to an albatross, a robin wing has a low aspect ratio and is therefore optimal for maneuvering the forest, but not optimal over the ocean. An albatross’s flight in a forest would be much less cost-efficient than it would be over the ocean, as it would expend significantly more energy struggling to get through the forest. So a high aspect ratio wing would be cost-efficient for certain behaviors in certain

habitats, a low aspect ratio wing would be cost-efficient for other behaviors and other habitats, etc. And if they do wing-propelled diving, there could be a third “most cost-efficient/optimal shape” for that behavior, accounting for flapping through the water, as discussed later. Each plot shows optimum shape space for very specific traits or behaviors, so authors should make sure the discussion text always reflects that sort of precise articulation of the context of things like “optimal.”

Line 232 – Good for including Pelagornis! I had written a thought that it should be mentioned somewhere, but deleted that thought, since I found it here.

Paragraph starting at line 298 – be very precise about what optimal means in this context; there are many different biological adverbs that could be added to change the meaning of the “optimals” throughout the paragraph – optimal for which aspect of a bird’s biology or behavior?

Line 567 – What does “characterized as outline shapes” entail? Wang and Clarke 2015 and Baumgart et al. 2021 use 2d landmarks to capture an outline. Which software was used to generate an outline of the wing? How was wing shape data collected from the outline?

Line 576 – “characterize seven six niches” – should pick a number

Reviewer #2

(Remarks to the Author)

In the manuscript “Bird wing planform shape is highly optimised for flight style”, the authors test the link between wing shape and flight style in birds using a new dataset and a creative morphometric approach. The authors created a theoretical morphospace onto which they have mapped flight “performance” metrics to create hypothetical performance spaces, which they paired with empirical shape measurements from a large number of avian taxa to test for functional optimization of wing morphology. The authors report some evidence for shape optimization, which is independent of phylogenetic effects. The manuscript is clear and well-written, and the subject will be of broad interest to the flight community. The figures, also, are very nice and well-presented. However, some aspects of the predictions and results remain unclear, and I do not believe that the authors have conducted the requisite tests to support their main claim. As such, I believe that some revision is necessary prior to publication. Below, I detail my major concerns with the study, as presented, followed by minor issues that should be addressed prior to publication.

Major concerns:

1) My first major concern with the study is that I do not find tests that adequately supports the main assertion of the manuscript, that “Wing planform shape is highly optimized for flight style”. My concern here is two-fold. First, I find the evidence for optimization to be mixed. The results seem to support high pitch agility across most of the studied species, and the species also cluster on a plateau of high second moment of area. Aspect ratio and tip angle show gradients, with one end approaching the putative “optima”, but the other end being far from it. In the composite performance surfaces, however, the species actually appear to sit within pareto valleys rather than being on peaks or ridges. I was rather hoping that the authors would discuss that, but they did not. My second concern is that while the authors did make an effort to quantify the proximity to the “optima” (through Euclidean distances between group means and the performance optima), I found no test that would discriminate between groups in their degree of optimization. Perhaps it would be worth calculating the optimization distance for each species and then doing a phylogenetic ANOVA (or similar) to show whether different groups (either taxonomic groups or behavioral/functional groups) share similar optimization ranges. The Euclidean distance metrics and any subsequent tests do need to be included in the methods (they are not at present) and presented in the results (not in the discussion as they are now).

2) My second major concern stems from the construction of the morphospaces and performance surfaces. The authors should be careful to note that they are, in most cases, using morphological proxies for performance rather than estimates of performance output (the exception here being the pitch agility metric – the rest of the “performance” surfaces are simply morphological traits). While the relationships between these morphological traits and their influence on flight performance are well-documented, it is unclear whether it is safe to assume a linear association between morphology and performance. I don’t have a problem with the use of these shape traits as proxies, but the authors should probably note this as a caveat of the study. Additionally, it is unclear how the authors arrived at the 20% figure to expand their theoretical grid space. This is an important consideration, because many performance traits do scale continuously (though not necessarily linearly) with shape – for example, the authors use aspect ratio as a proxy for flight efficiency. Indeed, wing aspect ratio is related to flight performance metrics like drag coefficient and lift-to-drag ratio, but the morphospace that the authors present appears to covary strongly with aspect ratio, so using it as a performance surface superimposed over the morphospace seems circular. It would be better to choose a true measure of flight performance (see Rader et al. 2020; Waldrop et al. 2020) rather than using the proxy.

3) I think that it would be helpful for the authors to set up more explicit predictions about how wing morphology is expected to vary with flight style / performance in the introduction, and then to follow a more rigorous hypothesis testing framework throughout the analyses. As the manuscript is currently presented, I struggled to understand what the authors were predicting, which made it difficult to ascertain whether the methodological approach was appropriate for the questions.

4) How was phylogenetic signal calculated for the dataset? In the methods, the authors list the `gm.prcmp` command, but in the results they talk about the `kmult` statistic (which does not appear in the methods).

Minor concerns:

1) Line 109: I'm not sure about "tongue-shaped" as a descriptor.

2) Double check spelling of references throughout.

- Line 157: "Wang and Clarke" is misspelled

- Line 389: "Rader and Hedrick" is misspelled

3) Some discussion of how the present results compare to 3-dimensional analyses (e.g., Rader and Hedrick 2023 and Rader et. al 2020) might be welcome.

4) Line 571: Define EFA.

5) Line 576: Typographic error; was it six or seven niches?

6) Line 638: Perhaps the authors could elaborate on how their performance metrics provide any "direct link to specific flight behaviors". This comment is probably best addressed by setting up some discrete predictions (see major concern #3).

7) Line 661: How are these planforms defined?

8) Line 688: I know that this equation is well-presented in the supplement of Harvey et al., but it would be helpful to define the terms here, especially as they pertain to the present data and methods.

References:

Rader JA, Hedrick TL, He Y, Waldrop LD. 2020. Functional Morphology of Gliding Flight II. Morphology Follows Predictions of Gliding Performance. *Integrative and Comparative Biology* 60:1297–1308.

Waldrop LD, He Y, Hedrick TL, Rader JA. 2020. Functional morphology of gliding flight I. Modeling reveals distinct performance landscapes based on soaring strategies. *Integrative and Comparative Biology* 60:1283–96.

Version 1:

Reviewer comments:

Reviewer #1

(Remarks to the Author)

I reviewed an earlier iteration of this paper before, and this version is stronger and does address concerns from reviewers. I think this is an acceptable version for publication.

One minor comment:

Line 85-86 – awkward phrasing, I'd suggest rearranging: "...represented on PC1 (53.75%) which describes the breadth of the wing chord, because all planforms are scaled to identical length." ("since" is a time word, "because" is causal.)

Reviewer #3

(Remarks to the Author)

Your innovative study of the planform of avian wings is fascinating and worthwhile. With one exception which I describe below, in general, I perceive that you revised your earlier manuscript in agreement with the recommendations of two reviewers. Specific to the major concerns of reviewer 2, your revisions effectively addressed concern 1 (Optimization is mixed), 3 (specific predictions and testing framework) and 4 (calculation of phylogenetic signal).

On the issue of using morphological proxies for performance (Reviewer 2 concern 2), I suggest that it is still not clear that your proxies are generally untested hypotheses about relationships between form and flight performance. You describe empirical properties and functional performance in ways that give the incorrect impression that there is strong evidence from prior studies when such evidence is lacking, particularly for your proxies of maneuverability and agility. Thus, I encourage you to add language throughout the manuscript that clarifies that your proxies are hypotheses that merit future empirical tests. Here are specific locations:

Line 11: Instead of "Empirical properties" use "Hypothesized empirical properties"

Line 13: Insert "hypothesized" before "functional performance and optimality"

Line 17: Insert "hypothesized to be" before "linked with efficient"

Line 62: Change wording for "established" Perhaps aspect ratio is adequately established as a proxy for cost of transport given various empirical studies, but the other proxies (2nd moment of area, agility index and tip angle) have not been adequately tested empirically as proxies for maneuverability, agility and induced power requirements, respectively.

Line 66: Insert "hypothesized" before "flight optima"

Line 78: Insert "hypothesized" before "functional optimality"

Line 103: Change header by inserting "Hypothesized" before "Functional Performance" This information is lacking on Line

Additional Comments:

In most cases when you use the word efficiency, it would be helpful to define the numerator and denominator, at least at first use. This might be lift:drag ratio, cost of transport (Joules per meter) or rotational efficiency (perhaps rotational energy per unit aerodynamic work). Specific instances include Line 17, Line 212, Line 232, Line 703, Line 747, Table 1 (first row below headers).

Maneuverability versus Agility: It would be helpful to define the difference between maneuverability and agility and clarify if there is any redundancy between Savile's and Harvey's predictions for the relationship between planform and the ability to vary body pitch. If the 2nd moment of area is large, the Savile (and later Rayner and Norberg) hypothesis seems to be that the aerodynamic surfaces are distributed relatively far from the axes of the body originating at the center of mass, so the wings can establish large rotational moments, presumably about the roll, pitch and yaw axes. The agility index of Harvey et al is specific to the ability to modulate pitch stability. Perhaps these proxies are not redundant if Savile's predictions are more for roll rather than pitch.

Many-to-One versus Pleiotropy (One-to-Many). I feel you or perhaps Reviewer 1 may have misinterpreted "many to one" being a potential explanation for why the planforms of birds occupy a subset of the potential morphospace. Many to one is a form of phenotypic redundancy, where many traits accomplish one performance. In the case of the Wainwright et al study you cite, the multibar linkage systems in the jaws of fish have many components that contribute to feeding. Redundancy is hypothesized to increase evolvability, because directional selection can act on one phenotypic trait while the necessary performance (e.g. feeding) is accomplished by other traits. I think the wings of birds are the reverse, where for many species one pair of wings must accomplish a variety of tasks (takeoff, cruising, gliding, maneuvering or even things like shading young from the sun or producing feather sounds (sonations) during sexually-selected displays or to alert conspecifics about danger from predators). This is therefore an example of one-to-many or pleiotropy, and it is hypothesized that one phenotype accomplishing many functions tends to constrain evolvability. A discussion of these ideas is available in:

Bergmann, P. J., & McElroy, E. J. (2014). Many-to-many mapping of phenotype to performance: an extension of the F-matrix for studying functional complexity. *Evolutionary Biology*, 41(4), 546-560.

also see:

Ghalambor, C. K., Walker, J. A., & Reznick, D. N. (2003). Multi-trait selection, adaptation, and constraints on the evolution of burst swimming performance. *Integrative and comparative biology*, 43(3), 431-438.

Figure 4: Consider using a different marker (e.g red dots) for the species in the performance groups you highlight at the top of each panel. For example, in panel 1, make all the marine soarer dots a different color than black so that it is easier to understand which species are near the optima.

Body Size: You mention that body size is a critical element not included in your metrics (Line 679). I suggest slightly more information would be helpful, and you might move some of this to the discussion. Yes, wing loading increases with increasing mass as you hint at on line 679, but also maximum mass-specific power available from the flight muscles is understood to decline as a function of increasing body mass, which is thought to explain why the smallest birds can easily out maneuver larger birds (e.g. mobbing behavior, aerial combat in hummingbirds, hawking in small passerines). Stated another way, most small birds are highly maneuverable and agile because they have marginal power to spare, and this may help explain why passerines, which are mostly on the small end of the range of body masses in birds, do not emerge from your study as having optimal planforms.

Signed: Bret Tobalske

REVIEWER COMMENTS

Reviewer #1 (Remarks to the Author):

The authors explore the range of existing avian wing shape relative to theoretical possibilities, a very fun question for the functional morphology community, and not often done. This is an important framework to think about how things have evolved the way they do and what may be guiding or constraining their direction of evolution. I made some comments below which I think could improve the manuscript, but I think it is going to be a good addition to the journal after revisions.

Line 80 – would this be better described as the wing aspect ratio? Based on the description in the next sentence, it sounds like PC1 is breadth of the wing chord relative to the wing length, not breadth of wing chord in isolation. Or is this a raw measurement of wing chord?

A phrase has been added here clarifying that all planforms, both theoretical and empirical are scaled to identical length. Due to this scaling the breadth of the chord, while functionally equal to aspect ratio, is better described as a raw measurement of the chord itself.

Line 138 – no suppl. fig number given

A supplementary figure number has been provided to fix this oversight.

Lines 144-146 – dominant by passerines or by non-waterbirds? I see passeriforms, but also cuculiforms, and galliforms on the far right. With all the other clades in that region that are hidden due to density of data points, I'd recommend being more inclusive with taxa listed in this contrasting statement, especially if non-passerine passeriforms are included in the analysis.

This passage has been rephrased to include Cuculiformes and Galliformes and to use the broader clade level designation Passeriformes rather than passerines, which previously suggested a more restricted class.

Line 156 – extra “/” included

The extra / has been removed

Line 157 – “Wang” instead of “Wange”

The spelling of Wang has been corrected

Line 162-163 – “burst flapping” should be hyphenated when used as an adjective

A hyphen has been added between burst and flapping

Line 164 – passerines or passeriforms? Taxonomic categories here are of the order level, so it may make more sense to refer to the group as passeriforms instead of passerines, which is a subset of Passeriformes.

Passeriforms has been substituted for passerines to in order to better clarify that we are referring to the entirety of the clade.

Thoughts on section at 166 – What about the idea of many-to-one mapping? (See Wainwright et al. 2005: <https://academic.oup.com/icb/article/45/2/256/778330>) Or the related idea of “this shape is sufficient for the task; taking the whole organism and its collection of needs and behaviors, it’s good enough.” Wings are not in isolation of the rest of the body, which would be a source for further constraints on empirical wing shape that could limit the amount of theoretical wing shape space that is achieved in reality.

This is an excellent point, that the rest of the animal is a likely source of constraint on the shape of the wing planform. While this will not affect the performance of the theoretical shapes, the elements tested by the analysis, it does warrant comment in reference to discrepancies in empirical wings. An acknowledgement of this and of many-to-one mapping has been added to the end of section a of the discussion.

Thoughts on sections at 185 – This is very interesting. Might differences in data collection protocols affect the differences found between the authors’ phylogenetic signal and that of Wang and Clarke 2015? They also captured the covert outline too. But it’s true that scale of analysis would also play a large effect.

A sentence has been added here acknowledging that the addition of covert feathering data by Wang and Clarke may also be a factor contributing to the discrepancy in phylogenetic signal between the datasets.

Line 217 (and throughout discussion) – Thinking about terms like “cost-efficient” and “optimal/non-optimal,” clear biological context matters with these sorts of statements. An albatross wing has a high aspect ratio and therefore optimal for soaring in an open marine environment, but it is not optimal for maneuvering in a forest. Relative to an albatross, a robin wing has a low aspect ratio and is therefore optimal for maneuvering the forest, but not optimal over the ocean. An albatross’s flight in a forest would be much less cost-efficient than it would be over the ocean, as it would expend significantly more energy struggling to get through the forest. So a high aspect ratio wing would be cost-efficient for certain behaviors in certain habitats, a low aspect ratio wing would be cost-efficient for other behaviors and other habitats, etc. And if they do wing-propelled diving, there could be a third “most cost-efficient/optimal shape” for that behavior, accounting for flapping through the water, as discussed later. Each plot shows optimum shape space for very specific traits or

behaviors, so authors should make sure the discussion text always reflects that sort of precise articulation of the context of things like “optimal.”

We agree that what is optimal in wings varies across bird groups and have made adjustments to the discussion to highlight that different groups of birds are hypothesised as optimal for specific flight styles. These hypotheses are made in reference to table 1 and are expanded upon in methods section c. additions have been made to methods section c to state which bird groups are hypothesised to be optimised for each flight style, linking table 1 into the text. We have also more explicitly outlined in the introduction that this is what underpins our analysis. We hope that this additional context improves the clarity of the underlying hypothesis that specific bird groups should experience functional pressure toward the development of optimally shaped wings for specific flight styles.

Line 232 – Good for including Pelagornis! I had written a thought that it should be mentioned somewhere, but deleted that thought, since I found it here.

We are glad to see that Pelagornis is a welcome addition, no change is necessary in response to this comment.

Paragraph starting at line 298 – be very precise about what optimal means in this context; there are many different biological adverbs that could be added to change the meaning of the “optimals” throughout the paragraph – optimal for which aspect of a bird’s biology or behavior?

The use of optimal to refer to both areas in morphospace and planform shape created some confusion., the paragraph has been restructured to clear up this confusion and specify when optimality refers to either the planforms or the areas of highest performance.

Line 567 – What does “characterized as outline shapes” entail? Wang and Clarke 2015 and Baumgart et al. 2021 use 2d landmarks to capture an outline. Which software was used to generate an outline of the wing? How was wing shape data collected from the outline?

Additional information about the initial capture of outline data and its digitisation has been added to methods section 2a including which programs were used for generating outline data from wing images.

Line 576 – “characterize seven six niches” – should pick a number

The number 7 has been removed to correct this oversight.

Reviewer #2 (Remarks to the Author):

In the manuscript “Bird wing planform shape is highly optimised for flight style”, the authors test the link between wing shape and flight style in birds using a new dataset and a creative morphometric approach. The authors created a theoretical morphospace onto which they have mapped flight “performance” metrics to create hypothetical performance spaces, which they paired with empirical shape measurements from a large number of avian taxa to test for functional optimization of wing morphology. The authors report some evidence for shape optimization, which is independent of phylogenetic effects. The manuscript is clear and well-written, and the subject will be of broad interest to the flight community. The figures, also, are very nice and well-presented. However, some aspects of the predictions and results remain unclear, and I do not believe that the authors have conducted the requisite tests to support their main claim. As such, I believe that some revision is necessary prior to publication. Below, I detail my major concerns with the study, as presented, followed by minor issues that should be addressed prior to publication.

Major concerns:

1) My first major concern with the study is that I do not find tests that adequately supports the main assertion of the manuscript, that “Wing planform shape is highly optimized for flight style”. My concern here is two-fold. First, I find the evidence for optimization to be mixed. The results seem to support high pitch agility across most of the studied species, and the species also cluster on a plateau of high second moment of area. Aspect ratio and tip angle show gradients, with one end approaching the putative “optima”, but the other end being far from it. In the composite performance surfaces, however, the species actually appear to sit within pareto valleys rather than being on peaks or ridges. I was rather hoping that the authors would discuss that, but they did not. My second concern is that while the authors did make an effort to quantify the proximity to the “optima” (through Euclidean distances between group means and the performance optima), I found no test that would discriminate between groups in their degree of optimization. Perhaps it would be worth calculating the optimization distance for each species and then doing a phylogenetic ANOVA (or similar) to show whether different groups (either taxonomic groups or behavioral/functional groups) share similar optimization ranges. The Euclidean distance metrics and any subsequent tests do need to be included in the methods (they are not at present) and presented in the results (not in the discussion as they are now).

To address the concern that the main finding of the paper, as expressed in the title, does not adequately match the findings, we have changed the title to more clearly state that the optimisation of avian wings for flight style is variable and not all groups exhibit wings with functionally optimal planform shapes. Since our analysis only tests the functional performance of theoretical shapes relative to each other, and not empirical shapes, there exists no discrete performance data for theoretical shapes

which could form the basis of between-group testing. We have emphasised the nature of this divide between theoretical and empirical forms to clarify this case.

2) My second major concern stems from the construction of the morphospaces and performance surfaces. The authors should be careful to note that they are, in most cases, using morphological proxies for performance rather than estimates of performance output (the exception here being the pitch agility metric – the rest of the “performance” surfaces are simply morphological traits). While the relationships between these morphological traits and their influence on flight performance are well-documented, it is unclear whether it is safe to assume a linear association between morphology and performance. I don’t have a problem with the use of these shape traits as proxies, but the authors should probably note this as a caveat of the study. Additionally, it is unclear how the authors arrived at the 20% figure to expand their theoretical grid space. This is an important consideration, because many performance traits do scale continuously (though not necessarily linearly) with shape – for example, the authors use aspect ratio as a proxy for flight efficiency. Indeed, wing aspect ratio is related to flight performance metrics like drag coefficient and lift-to-drag ratio, but the morphospace that the authors present appears to covary strongly with aspect ratio, so using it as a performance surface superimposed over the morphospace seems circular. It would be better to choose a true measure of flight performance (see Rader et al. 2020; Waldrop et al. 2020) rather than using the proxy.

The introduction has been modified to express that the performance metrics tested represent proxies for performance based on established understanding of the relationship between morphology and function. These proxies do not represent a true measure of performance output, and this is why we only test theoretical shapes, rather than the empirical planforms. This is the benefit of theoretical morphology as it allows us to produce analyses of relative functional performance independent of attempting to ascertain the discrete performance of individual taxa. To avoid confusion, we have adjusted the phrasing of the introduction to better emphasise this divide and that only theoretical shapes are tested with the functional proxies. Additionally, the 20% figure for expanding morphospace is determined by the limits of theoretical shape generation, producing theoretical shapes what are possible and not too far from winglike shapes. It would hypothetically be possible to expand this space indefinitely to include winglike shapes as well as simple polygons etc. though in practice this would be unnecessarily computationally expensive and include a vast number of self-intersecting forms which are impossible and therefore untestable. We have added clarification of this point to section b of the methods to clarify this point.

3) I think that it would be helpful for the authors to set up more explicit predictions about how wing morphology is expected to vary with flight style / performance in the introduction, and then to follow a more rigorous hypothesis testing framework throughout the analyses. As the manuscript is currently presented, I struggled to

understand what the authors were predicting, which made it difficult to ascertain whether the methodological approach was appropriate for the questions.

The introduction has been expanded to include reference to the underlying hypothesis that specific bird groups should possess optimal planforms for specific metric combinations, with reference to table 1 which outlines the flight style groups analysed and their link to specific metric combinations. This was previously underemphasised within the introduction and hopefully has now been reinforced.

4) How was phylogenetic signal calculated for the dataset? In the methods, the authors list the gm.prcmp command, but in the results they talk about the kmult statistic (which does not appear in the methods).

The section of methods addressing the calculation of phylogenetic signal has been amended to draw distinction between the commands used for phylogenetic signal, which generate a multivariate k statistic (kmult) and the command used for ancestral shape reconstruction, gm.prcmp.

Minor concerns:

1) Line 109: I'm not sure about "tongue-shaped" as a descriptor.

The term tongue-shaped has been replaced with "a rectangular region of optimality" to improve specificity in description of the shape of the optimal peak.

2) Double check spelling of references throughout.

- Line 157: "Wang and Clarke" is misspelled

- Line 389: "Rader and Hedrick" is misspelled

These misspellings have been corrected

3) Some discussion of how the present results compare to 3-dimensional analyses (e.g., Rader and Hedrick 2023 and Rader et. al 2020) might be welcome.

Reference has been added to the existence of these 3-dimensional studies in the introduction and comparison of our results to those of Rader et al. 2020 has been added to section c of the discussion.

4) Line 571: Define EFA.

This acronym has been expanded to Elliptical Fourier Analysis.

5) Line 576: Typographic error; was it six or seven niches?

The extra number has been removed.

6) Line 638: *Perhaps the authors could elaborate on how their performance metrics provide any “direct link to specific flight behaviors”. This comment is probably best addressed by setting up some discrete predictions (see major concern #3).*

Along with modifications to the discussion and introduction, the discussion section c had been modified to make more explicit which bird groups each of the four functional metrics are hypothesised to be optimising for, with reference made to table 1. We hope that the addition of these predictions into the main text further clarifies what we mean when we say that a bird group should be optimal a particular metric or combination of metrics.

7) Line 661: *How are these planforms defined?*

Additional clarification of what we mean by planform and how the information was captured has been added to section 2a of the methodology. This should hopefully explain that planform here represents the outline of the righthand wing when viewed from directly above.

8) Line 688: *I know that this equation is well-presented in the supplement of Harvey et al., but it would be helpful to define the terms here, especially as they pertain to the present data and methods.*

To better explain how the formula derived by Harvey et al. relates to our theoretical planforms, additional clarification of terms has been added to section d of methods, with explanation of how these parts were calculated for theoretical outline shapes.

References:

Rader JA, Hedrick TL, He Y, Waldrop LD. 2020. Functional Morphology of Gliding Flight II. Morphology Follows Predictions of Gliding Performance. Integrative and Comparative Biology 60:1297–1308.

Waldrop LD, He Y, Hedrick TL, Rader JA. 2020. Functional morphology of gliding flight I. Modeling reveals distinct performance landscapes based on soaring strategies. Integrative and Comparative Biology 60:1283–96.

REVIEWER COMMENTS

Reviewer #1 (Remarks to the Author):

One minor comment:

Line 85-86 – awkward phrasing, I'd suggest rearranging: "...represented on PC1 (53.75%) which describes the breadth of the wing chord, because all planforms are scaled to identical length." ("since" is a time word, "because" is causal.)

This sentence has been rewritten to fit the reviewers improved phrasing and to replace 'since' with 'because'.

Reviewer #3 (Remarks to the Author):

On the issue of using morphological proxies for performance (Reviewer 2 concern 2), I suggest that it is still not clear that your proxies are generally untested hypotheses about relationships between form and flight performance. You describe empirical properties and functional performance in ways that give the incorrect impression that there is strong evidence from prior studies when such evidence is lacking, particularly for your proxies of maneuverability and agility. Thus, I encourage you to add language throughout the manuscript that clarifies that your proxies are hypotheses that merit future empirical tests. Here are specific locations:

Line 11: Instead of "Empirical properties" use "Hypothesized empirical properties"

Line 13: Insert "hypothesized" before "functional performance and optimality"

Line 17: Insert "hypothesized to be" before "linked with efficient"

Line 62: Change wording for "established" Perhaps aspect ratio is adequately established as a proxy for cost of transport given various empirical studies, but the other proxies (2nd moment of area, agility index and tip angle have not been adequately tested empirically as proxies for maneuverability, agility and induced power requirements, respectively.

Line 66: Insert "hypothesized" before "flight optima"

Line 78: Insert "hypothesized" before "functional optimality"

Line 103: Change header by inserting "Hypothesized" before "Functional Performance" This information is lacking on Line 17,

The suggested insertions have been made to better reflect the degree of clarity in the scientific understanding of the link between the empirical properties tested in this analysis and their link to flight performance. Additionally, the word 'established' in line 62 has been replaced with 'proposed' to further emphasise this point.

In most cases when you use the word efficiency, it would be helpful to define the numerator and denominator, at least at first use. This might be lift:drag ratio, cost of transport (Joules per meter) or rotational efficiency (perhaps rotational energy per unit aerodynamic work). Specific instances include Line 17, Line 212, Line 232, Line 703, Line 747, Table 1 (first row below headers).

The manuscript has been edited to replace the term 'efficiency' with the more relevant descriptor of low transport cost flight. This should hopefully reduce confusion about the nature of the metric used as higher aspect ratio is a proxy for energy savings through drag reduction. The addition of units for efficiency is not relevant to this research as efficiency is not the metric being tested.

Maneuverability versus Agility: It would be helpful to define the difference between maneuverability and agility and clarify if there is any redundancy between Savile's and Harvey's predictions for the relationship between planform and the ability to vary body pitch. If the 2nd moment of area is large, the Savile (and later Rayner and Norberg) hypothesis seems to be that the aerodynamic surfaces are distributed relatively far from the axes of the body originating at the center of mass, so the wings can establish large rotational moments, presumably about the roll, pitch and yaw axes. The agility index of Harvey et al is specific to the ability to modulate pitch stability. Perhaps these proxies are not redundant if Savile's predictions are more for roll rather than pitch.

The methods section of the paper has been altered to better reflect the differences between agility and manoeuvrability, emphasising that these metrics describe movement on different axes with agility a function of pitch and manoeuvrability yaw. This demonstrates that there is no redundancy between these metrics and, while both characterise change in direction in space, they refer to different specific aspects of this movement.

Many-to-One versus Pleiotropy (One-to-Many). I feel you or perhaps Reviewer 1 may have misinterpreted "many to one" being a potential explanation for why the planforms of birds occupy a subset of the potential morphospace. Many to one is a form of phenotypic redundancy, where many traits accomplish one performance. In the case of the Wainwright et al study you cite, the multibar linkage systems in the jaws of fish have many components that contribute to feeding. Redundancy is hypothesized to increase evolvability, because directional selection can act on one phenotypic trait while the necessary performance (e.g. feeding) is accomplished by other traits. I think the wings of birds are the reverse, where for many species one pair of wings must accomplish a variety of tasks (takeoff, cruising, gliding,

maneuvering or even things like shading young from the sun or producing feather sounds (sonations) during sexually-selected displays or to alert conspecifics about danger from predators). This is therefore an example of one-to-many or pleiotropy, and it is hypothesized that one phenotype accomplishing many functions tends to constrain evolvability. A discussion of these ideas is available in:

Bergmann, P. J., & McElroy, E. J. (2014). Many-to-many mapping of phenotype to performance: an extension of the F-matrix for studying functional complexity. Evolutionary Biology, 41(4), 546-560.

also see:

Ghalambor, C. K., Walker, J. A., & Reznick, D. N. (2003). Multi-trait selection, adaptation, and constraints on the evolution of burst swimming performance. Integrative and comparative biology, 43(3), 431-438.

Additional text has been added to the end of discussion section 3a address the possibility of pleiotropy as a constraint on bird wing shape. This form of constraint is likely one of many factors which combine to drive the differential evolution of optimal wings across the dataset, and its inclusion makes for a more comprehensive discussion of flight constraints. Additionally the two proposed references have been added to the citation list.

Figure 4: Consider using a different marker (e.g red dots) for the species in the performance groups you highlight at the top of each panel. For example, in panel 1, make all the marine soarer dots a different color than black so that it is easier to understand which species are near the optima.

Body Size: You mention that body size is a critical element not included in your metrics (Line 679). I suggest slightly more information would be helpful, and you might move some of this to the discussion. Yes, wing loading increases with increasing mass as you hint at on line 679, but also maximum mass-specific power available from the flight muscles is understood to decline as a function of increasing body mass, which is thought to explain why the smallest birds can easily out maneuver larger birds (e.g. mobbing behavior, aerial combat in hummingbirds, hawking in small passerines). Stated another way, most small birds are highly maneuverable and agile because they have marginal power to spare, and this may help explain why passerines, which are mostly on the small end of the range of body masses in birds, do not emerge from your study as having optimal planforms.

Three additional sentences have been added to discussion section f to incorporate this idea. The relaxation of constraint as a result of proportionally greater muscle power may help to explain the lack of optimality in passerine birds but critically is not a universal trend as hummingbirds possess optimal planforms and appear to be highly constrained by function despite their size. The text of the article has been edited to reflect this.